# Preterm infants' first breastfeeding attempt: Early initiation and performance: A large multicentre questionnaire study based on maternal observations

Ragnhild Maastrup[1,2]*, Sisse Walloee[3], Hanne Kronborg[4], Helle B. Sandfeld[5],
Ane L. Rom[2,6,7]

**1** Department of Neonatology, Knowledge Centre for Breastfeeding Infants with Special Needs,
Copenhagen University Hospital Rigshospitalet, Copenhagen, Denmark, **2** Research Unit Women's and
Children's Health, Juliane Marie Centre, Copenhagen University Hospital Rigshospitalet, Copenhagen,
Denmark, **3** Dept of Clinical Research, OPEN—Patient data Explorative Network, University of Southern
Denmark, Odense, Denmark, **4** Department of Public Health, Section for Nursing, Aarhus University,
Aarhus, Denmark, **5** Department of Paediatrics, Region Hospital Randers, Randers, Denmark,
**6** Department of Obstetrics, The Juliane Marie Centre, Copenhagen University Hospital Rigshospitalet,
Copenhagen, Denmark, **7** Research Unit of Gynaecology and Obstetrics, Department of Clinical
Research, University of Southern Denmark, Odense, Denmark

* ragnhild.maastrup@regionh.dk

journal.pone.0303224

Beirut Medical Center, LEBANON

**Peer Review History:** PLOS recognizes the
benefits of transparency in the peer review
process; therefore, we enable the publication
of all of the content of peer review and
author responses alongside final, published
articles. The editorial history of this article is
available here: https://doi.org/10.1371/journal.
pone.0303224

## Abstract

The Baby-friendly Hospital Initiative for neonatal wards and the World Health Orga-
nization recommend that stable preterm infants initiate breastfeeding regardless of
gestational age, postmenstrual age (PMA), or weight. Documented practice, how-
ever, is limited. We aimed to describe PMA at first breastfeeding attempt of stable
preterm infants, to analyse delaying factors, to detect differences in breastfeeding
performance across gestational age groups and use of nasal-CPAP. This Danish
multicentre cohort study was based on questionnaires answered by mothers of
992 preterm infants gestational age 23–36 weeks. Differences in PMA between
gestational age groups at first breastfeeding attempt were analysed by One-way
ANOVA, and associations between PMA and selected factors by linear regression
models. The lowest PMA at first breastfeeding attempt was 27.57 weeks. Of the
extremely and very preterm infants, 61% and 46%, respectively, had the first breast-
feeding attempt before PMA 32 weeks. Mechanical ventilation significantly delayed
first breastfeeding attempt by seven days ($p < 0.0001$). Performance at the preterm
infants' first breastfeeding attempt were predominantly without swallowing (78%).
During first attempt, 29% were at breast with nasal-CPAP. Performance was in
general not affected by nasal-CPAP treatment. In this cohort of preterm infants, we
conclude that early initiation of breastfeeding is possible, also at low PMA and while
maintained on nasal-CPAP. Hence, nasal-CPAP should not be a barrier for breast-
feeding initiation. At first breastfeeding, even preterm infants before PMA 32 weeks

**Data availability statement:** The authors confirm that, the data are available on Figshare https://figshare.com/articles/dataset/Preterm_infants_first_breastfeeding/24049401.

**Funding:** The authors disclosed receipt of the following financial support for the research and authorship of this article: Funding to support the research was received from: Novo Nordisk Foundation, grant number NNF15OC0018156 and NNF17OC0030080 https://novonordiskfonden.dk/en/ (RM), Lundbeck Foundation, grant number F-23137-01 https://www.lundbeckfonden.com/ (RM), Helsefonden, grant number 16-B-0059 https://helsefonden.dk/ (RM), Department of Neonatology, Rigshospitalet, Copenhagen University Hospital https://www.rigshospitalet.dk/afdelinger-og-klinikker/julianemarie/intensiv-behandling-af-nyfoedte-og-mindre-boern/Sider/default.aspx (RM), and Rigshospitalet, Copenhagen University Hospital, Research Committee, grant number E-23137-01 https://www.rigshospitalet.dk/ (RM). The funders had no role in study design, data collection and analysis, decision to publish, or preparation of the manuscript.

**Competing interests:** The authors have declared that no competing interests exist.

demonstrated breastfeeding behaviours, although the majority did not swallow. Preterm infants need time to familiarize with the breast.

## Introduction

### Timing of breastfeeding initiation

Since 2013, the Baby-friendly Hospital Initiative for neonatal wards (Neo-BFHI) has recommended that stable preterm infants should be offered unrestricted access to the breast regardless of gestational age (GA), postmenstrual age (PMA), post-natal age (PNA), or current weight [1,2], and in 2020, The World Health Organization (WHO) repeated this recommendation [3]. Twenty years ago, evidence from Sweden showed that preterm infants have early breastfeeding competencies and that they manage to grasp nipple and areola and suckle the breast from PMA 27.9 weeks [4]. A Danish study found that extremely preterm infants initiate breastfeeding at a mean PMA of 31.8 weeks [5] and two studies from the US described first breastfeeding experiences from PMA 30 and 31 weeks, respectively [6,7]. A systematic review suggested that stable preterm infants could safely be exposed to the breast before PMA 32 weeks [8]. Finally, Canadian research showed breastfeeding initiated from 30.1 weeks PMA among very preterm infants, and of these 36% had the first attempt while maintained on nasal Continuous Positive Airway Pressure (CPAP), without registering any adverse events [9]. However, many studies from Europe, Australia and the U.S. describe breastfeeding initiation from 32 or 34 weeks PMA [10–12], and guidelines and policies in many countries still state that breastfeeding should not be initiated before a specific PMA, often between 32 and 34 weeks [13–15].

The delay of the timing of first breastfeeding attempt in preterm infants may be influenced by severe conditions of the infant, the small size of the mouth in the more immature infant, and lack of breastfeeding experience in the mother. Also, treatment with nasal-CPAP may influence the timing. Danish and Canadian studies found that 21% and 36% of preterm infants, respectively, were offered the breast during treatment with nasal-CPAP [5,9]. These studies, however, did not assess potential delaying factors.

### Breastfeeding performance

Ideally, the breastfeeding process begins with skin-to-skin contact, and after activity and crawling movements, the infant locates the breast, licks and tastes and familiarizes with the nipple, latches, and then breastfeeds, which is referred to as "the nine stages" [16]. Term infants establish breastfeeding within few hours following birth. Even though preterm infants have a longer journey to exclusive breastfeeding that can last for days, weeks, or months [4], it includes a pattern similar to infants born at term. The breastfeeding journey for preterm infants was previously described in the so-called 'Milky Way', from skin-to-skin contact and developing step by step to exclusive breastfeeding (Fig 1).

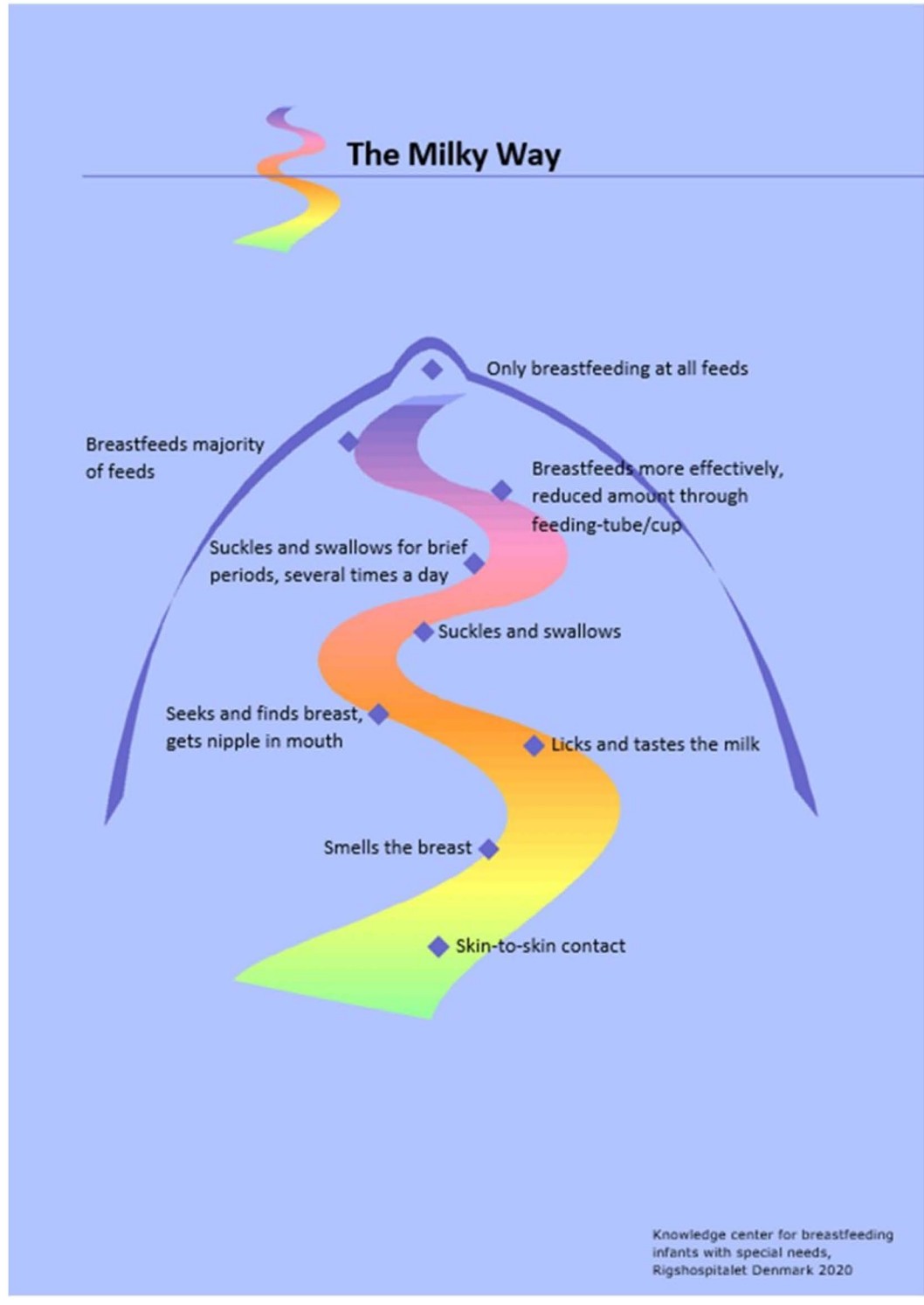

**Fig 1. The Milky Way, the breastfeeding journey for preterm infants.**

   

The Milky Way was developed by our group in Denmark in 2002 (revised 2020) inspired by Berlith Persson's Breast-feeding Wheel [17,18]. In a Swedish study, Nyqvist et al. described that the majority of preterm infants rooted at their first breastfeeding attempt, that a few only licked and tasted, that all had the nipple in the mouth, and that most sucked occasionally or repeatedly [4]. They included, however, only healthy singleton preterm infants without nasal-CPAP. Studies assessing the first breastfeeding performance in a general population of preterm infants, including infants with respiratory support remain sparse.

### Research gap and aim

Most of the existing studies of PMA and performance at first breastfeeding attempt in preterm infants have been limited by size (15–96 participants), by lack of information on infant criteria for initiating breastfeeding or by lack of analyses on potential delaying factors [4–7,9]. As an exception, one study described that the infants had to be without respiratory support to initiate breastfeeding [4] and two studies that breastfeeding could be initiated with nasal-CPAP [5,9]. Therefore, using Neo-BFHI's and WHO's recommendations for initiation of breastfeeding in preterm infants, we aimed to assess stable preterm infants' first breastfeeding attempt by, i.e., 1) PMA and PNA at first attempt according to GA, 2) delaying factors associated with PMA at first attempt, 3) performance at first attempt with and without nasal-CPAP, and 4) the correlation between PMA at first attempt and PMA at exclusive breastfeeding establishment as well as exclusive breastfeeding at discharge.

## Materials and methods

### Ethic statement

The mothers of the preterm infants were informed that participation in the study was voluntary. Each participating mother signed a written consent. The study was conducted in accordance with the Declaration of Helsinki [19] and approved by The Danish Data Protection Agency (Journal number 2012-58-0004, RH-2016–321, I-Suite 05022). Since it will always be the mother's individual decision whether she breastfeeds, it was highlighted to the neonatal nurses that breastfeeding guidance should focus on supporting mothers in reaching their personal goals for breastfeeding.

### Design and setting

An observational multicenter cohort study with retrospective data was conducted from 2016–2019. All 17 Danish NICUs and one children's department, which routinely cared for preterm infants during breastfeeding establishment, were invited to participate in the multicentre study. As all wards provided respiratory support for newborn infants (nasal-CPAP or mechanical ventilation) they are for practical reasons all referred to as NICUs. Of the 18 NICUs, 13 participated in the study. Reasons for not participating were other ongoing breastfeeding studies (three NICUs) and lack of time for breast-feeding research (two NICUs). The 13 participating NICUs had a mean of 18.7 beds.

Health care in Denmark is free of charge. Preterm infants born before 35 gestational weeks are admitted to a Danish NICU; in most hospitals, infants born ≥35 gestational weeks are admitted to a NICU only if they need additional treatment and care. Family-centered care is common in Danish NICUs [20]. Most of the Danish NICUs have an early discharge program where infants transit from tube-feeding to breastfeeding at home with support from the NICU at least twice a week. For infants in these programs, discharge is regarded as discharge from the program. A core element of neonatal nursing in Denmark is supporting mothers in early initiation and establishment of breastfeeding in small, sick, and preterm infants. Common practice in Danish NICU is that infant stability is the criterion for early breastfeeding initiation, thus following the Neo-BFHI and WHO recommendation, that stable preterm infants should be offered unrestricted access to the breast regardless of GA, PMA, PNA, or current weight [1–3]. NICUs in Denmark comply with the WHO and Neo-BFHI definition of infant stability related to breastfeeding: "Infants who respond to routine care and handling

without experiencing severe apnoea, desaturation and bradycardia" including lack of significant blood pressure fluctuations [1–3]. Infant stability will be assessed by the competent neonatal nurses. Even if nasal-CPAP is not an obstacle for breastfeeding initiation, the practice is under evolution in Denmark. For those infants only treated with nasal-CPAP for a few hours, feeding cues might not appear when the mother is present. Further, some of the extremely end very preterm infants could be stable for days before the first breastfeeding attempt depending on the individual nurse. In general, preterm infants are not discharged from the NICU or from the early discharge program until exclusive breastfeeding is established or given up by the mother and other feeding methods have been established. The NICUs have mandatory breastfeeding education for new nurses. Mechanical ventilation is used less than nasal-CPAP in Denmark for treatment of preterm infants with respiratory distress syndrome [21]. In 2016–2018 high-flow nasal cannula was not used frequently. No hospitals in Denmark hold a valid BFHI designation as the program was closed in Denmark in 2008. About 97–99% of Danish term and preterm infants initiate breastfeeding [5,22], and 38% of preterm infants and 60% of term infants are exclusively breastfed for four months [5,23].

## Participants

The study combined data from two comparable cohorts of preterm mother-infant dyads born from October 1, 2016 to July 31, 2017 and from February 1 to December 31, 2018. The two cohorts were part of a pre-post staff training intervention study improving exclusive breastfeeding in NICUs [24]. First breastfeeding attempt was not part of the intervention.

Inclusion criteria for participating in the study were preterm infants (<37 gestational weeks) admitted to the NICU within the first five days of birth. Exclusion criteria were infants discharged from NICU to maternity or paediatric wards other than neonatal, mothers of preterm infants with history of drug abuse (which precluded the recommendation to breastfeed), and mothers not able to read the questionnaires in Danish even with help from the family. Mothers who were particularly vulnerable were not invited to participate in the study, e.g., giving infant away for foster care, psychiatric problems, or extraordinarily affected by the infant's severe conditions.

## Data collection

The present study used two questionnaires for mothers of preterm infants that, with few revisions, were adapted from a Danish study of breastfeeding preterm infants [5,25]. The revision included adding a question about infant performance at first breastfeeding attempt and deleting questions about breastfeeding experiences in the mother's network. The questionnaires were pilot tested by one mother for combability of the online platform. The first questionnaire obtained information about demographic data on infant (e.g., GA, birth weight, gender) and mother (e.g., age, education), mode of delivery, previous breastfeeding experience and smoking status. Information about date and performance of the first breastfeeding attempt were also collected in the first questionnaire (if occurred before answering the first questionnaire) and repeated in the second questionnaire including the option "I have answered this before". If mothers nevertheless answered this question twice, the first response was chosen because of the shortest recall period. The second questionnaire obtained Information about breastfeeding at discharge (exclusive, partial, or non), use of bottle-feeding, and infant respiratory support. The questionnaires are further described in a previous paper [24].

The questionnaires were answered in an online platform (EasyTrial). The mother received a link by e-mail to the first questionnaire approximately one week after delivery, and a link to the second at the infant's final discharge to home. Reminders were sent alternately by text message and e-mail at least three times in the following month.

In each participating NICU, two or more contact nurses were appointed to register all preterm infants admitted to the NICU during the study periods and inform and enrol mother-infant dyads in the study. To improve adherence to the study, supportive and encouraging e-mails were sent to the contact nurses and nurse managers at two to three months intervals. The principal investigator was available to support the contact nurses and respond to questions.

## Variables

Outcome variables were timing of and performance at first breastfeeding attempt. Timing of the event was measured by PMA in weeks and PNA in days. PMA was defined as the time elapsed between the first day of the last menstrual period and birth plus the time elapsed from birth to the outcome of interest. PNA was defined as time elapsed from birth to the outcome of interest. Performance was categorized according to the steps on the Milky Way (Fig 1). The Milky Way has been used nationwide in Denmark since 2008 to inform parents and health care professionals about preterm breastfeeding [26]. Mothers of preterm infants were therefore likely to be familiar with the steps in the Milky Way. The mothers were asked to choose the infant's best performance on the Milky Way at the first breastfeeding attempt. "Smells the breast" is the step on the Milky Way that might not include activity from the infant and could also be described as "Mouth/nose against nipple". The first breastfeeding attempt was defined as the first time the infant was placed in front of the mother's nipple and had the possibility for sucking at the breast, no matter how much breastfeeding activity the infant demonstrated. The day of the first breastfeeding attempt was defined as the day of breastfeeding initiation.

Factors that could be associated with timing of first breastfeeding attempt were: GA (weeks), maintained on nasal-CPAP or high flow nasal cannula (both categorized as nasal-CPAP), mechanical ventilation, small for gestational age, and lack of previous breastfeeding experience (all categorized yes/no).

Characteristics of infants and mothers included gender (boy/girl), mode of delivery (vaginal or caesarean section), education (low or none: < 14 years, intermediate: 14–16 years, high: > 16 years of education), breastfeeding experience (yes/no), maternal age more than 30 years (yes/no), and current smoker (yes/no).

As GA and birth weight are highly correlated, GA was selected in favour of birth weight and additional influence of birth weight was measured by small for gestational age. Severe condition of the infant was measured by mechanical ventilation. Establishment of exclusive breastfeeding was dichotomized and defined as exclusive directly breastfeeding. Timing of first bottle-feed was answered by the mother if it occurred in the NICU, and this information was missing for those initiating bottle-feeding during the early discharge program.

## Statistics

Characteristics of infants and mothers, distribution of infants initiating breastfeeding within different weeks of PMA, and distribution of performance at first breastfeed were assessed descriptively by numbers and percentages. Distribution of PMA at first breastfeeding attempt across GA groups were reported by means and standard deviations (SD) and PNA by median and interquartile range (IQR). To determine statistically significant differences in the normally distributed data of PMA between GA groups at first breastfeeding attempt One-way ANOVA was applied. Differences in non-normally distributed data of PNA at first breastfeeding attempt were tested with Kruskal-Wallis test.

The association between selected potentially delaying factors and PMA at the first breastfeeding attempt was analysed by linear regression models. Potentially delaying factors included treatment with nasal-CPAP, mechanical ventilation, small for gestational age, and lack of previous breastfeeding experience, and all were independently adjusted for GA, and in the final model adjusted for all the included factors. The regression analysis was carried out with one infant per mother to ensure that mothers of twins and triplets were only counted once.

Associations between performance and GA, PMA groups, and in infants with and without nasal-CPAP were reported with Pearson Chi-Square and Linear-by-Linear Association. In sub analyses of nasal-CPAP, breastfeeding performance was dichotomised to breastfeeding behaviour without swallowing (including "Smells the breast", "Licks and tastes the milk", and "Seeks and finds breast, gets nipple in mouth") and breastfeeding behaviour with swallowing (including "Suckles and swallows briefly", "Breastfeeds more effectively, reduced amount through feeding-tube/cup, "Breastfeeds majority of the feed", and "Breastfeeds a complete feed").

To test the correlation between PMA at first breastfeeding attempt and at establishment of exclusive breastfeeding, Pearson correlation coefficient was calculated. The association between PMA at first breastfeeding attempt and exclusive breastfeeding at discharge was analysed by a linear regression model in each GA group.

Some mothers did not answer all of the questions or did not answer the second questionnaire about breastfeeding at discharge. Mothers and infants with missing data were excluded from the respective analyses. SPSS version 25 was used for statistical analyses. Values of p < 0.05 were considered statistically significant.

## Results

In total 1819 preterm infants were eligible. Mothers of 125 infants were particularly vulnerable (e.g., psychiatric problems, extremely stressed by their critically ill infant) and not approached for inclusion, further, the NICUs randomly forgot to include 174 infants. Mothers of 1520 infants were informed and 277 declined to participate. Consent for participation was obtained for 1243 infants and 992 (79.8%) of them had available data on performance at first breastfeeding attempt (see Flowchart, Fig 2).

Table 1 shows the characteristics of the 992 infants and 858 mothers in the study population.

The largest group were the moderate preterm infants (GA 32–34 weeks), and 29% of the mothers had breastfed before (Table 1). Nasal-CPAP treatment without any additional mechanical ventilation was used in 62.9% of the preterm infants. In total, 29.4% of the preterm infants were maintained on nasal-CPAP during the first breastfeeding attempt (Table 1). The lower the GA group the more infants were maintained on nasal-CPAP at the first breastfeeding attempt from 58/69 (84.1%) in extremely preterm infants to 59/273 (21.6%) in late preterm infants (Linear-by-Linear Association <0.0001, see full table in Supporting information S1 Table). Further, we found significant differences between NICUs from 11.2% to 43.8% of infants maintained on nasal-CPAP during first breastfeeding attempt (p < 0.0001).

## Flowchart

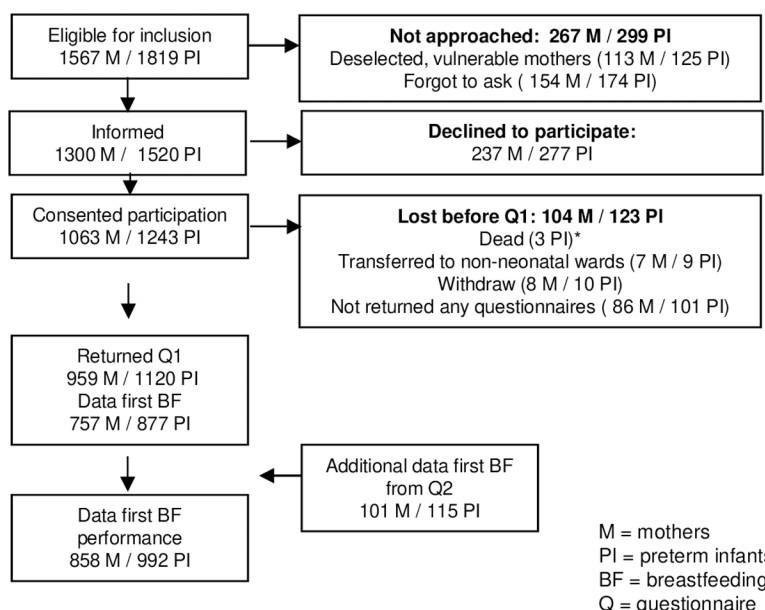

**Fig 2. Flowchart.**

**Table 1. Characteristics of participating infants and mothers.**

| | N 992 | | Missing |
|---|---|---|---|
| | n/N | % | n |
| **Infant** | | | |
| Gestational age <28 | 69/992 | 7.0 | 0 |
| Gestational age 28–31 | 210/992 | 21.2 | 0 |
| Gestational age 32–34 | 440/992 | 44.3 | 0 |
| Gestational age 35–36 | 273/992 | 27.5 | 0 |
| Small for gestational age | 182/982 | 18.5 | 10 |
| Multiple birth | 272/992 | 27.4 | 0 |
| Gender, boys | 600/992 | 60.5 | 0 |
| Mechanical ventilation (MV) followed by nasal-CPAP | 66/876 | 7.5 | 116 |
| Mechanical ventilation (MV) not followed by nasal-CPAP | 3/876 | 0.3 | 116 |
| Nasal-CPAP or high flow nasal cannula (not MV) | 551/876 | 62.9 | 116 |
| Nasal-CPAP during first breastfeeding attempt | 292/992 | 29.4 | 0 |
| Oxygen (with or without CPAP) during first breastfeeding attempt | 158/992 | 15.9 | 0 |
| **Mother** | | | |
| Breastfed before | 246/846 | 29.1 | 12 |
| Mode of delivery, caesarean section | 455/848 | 53.7 | 10 |
| Maternal age more than 30 years | 387/838 | 46.2 | 20 |
| Education, low or none (<14 years) | 176/838 | 21.0 | 20 |
| Education, intermediate (14–16 years) | 431/838 | 51.4 | 20 |
| Education, high (>16 years) | 231/838 | 27.6 | 20 |
| Smoking | 43/838 | 5.1 | 20 |

CPAP = Continuous positive airway pressure, MV = Mechanical ventilation.

Most infants had the first breastfeeding attempt before the first questionnaire was answered with a maternal recall period from the day of first breastfeeding attempt to answering the questionnaire of a median of 9 days (IQR 5–14) from the 741 mothers. Ninety-eight mothers (12%) answered this question in the second questionnaire with a recall period of median 52 days (IQR 14–85).

## Timing of first breastfeeding attempt

The lowest PMA at first breastfeeding attempt was 27.57 weeks. There were significant differences in PMA at first breastfeeding attempt across the four different GA groups (Table 2).

Of the extremely preterm infants (GA<28 weeks), 38.5% initiated breastfeeding before 30 weeks PMA, and 60.0% before 32 weeks PMA. Of the very preterm infants (GA 28–31), 46.5% initiated breastfeeding before 32 weeks PMA (Table 3). Of the 487 infants born before 34 weeks GA, 395 (81.1%) initiated breastfeeding before 34 weeks PMA (See Scatterplot, Supporting information S2 Fig).

## Factors associated with PMA at first breastfeeding attempt

In the analyses of the association between selected potentially delaying factors and PMA at first breastfeeding attempt, we found, after adjusting for GA, that infants who were mechanically ventilated had the first breastfeeding attempt significantly later compared to those not mechanically ventilated (6.93 days (95% CI 4.20 to 9.65), p<0.0001) (Table 4). In contrast, being treated with nasal-CPAP, being small for gestational age, or having a mother without previous breastfeeding

**Table 2. Postmenstrual age at first breastfeeding attempt across gestational age groups.**

| | GA<28 | GA 28–31 | GA 32–34 | GA 35–36 | P-value |
|---|---|---|---|---|---|
| N=967 | 65 | 202 | 427 | 273 | |
| PMA weeks, mean (SD) | 31.33 (2.69) | 32.14 (1.39) | 34.16 (0.91) | 35.98 (0.61) | <0.001 |
| 95% CI for PMA mean | 30.67 - 32.00 | 31.94 - 32.33 | 34.07 - 34.25 | 35.91 - 36.05 | |
| PMA Range | 27.57 - 40.71 | 29.14 - 38.57 | 32.14 - 43.14 | 35.00 - 38.66 | |
| PNA days, median (IQR) | 32 (16 - 47) | 8 (5 - 14) | 2 (1 - 4) | 1 (0 - 2) | <0.001 |
| | N=58 | N=187 | N=386 | N=253 | |
| Weight, mean grams (SD) | 1368 (457) | 1570 (405) | 2098 (430) | 2460 (493) | <0.001 |

GA=gestational age, PMA=postmenstrual age, PNA=postnatal age, SD=standard deviation, IQR=interquartile range.

**Table 3. Distribution of preterm infants' first breastfeeding attempt within different weeks of postmenstrual age.**

| Weeks PMA | N | <28 n/N (%) | 28–<30 n/N (%) | 30–<32 n/N (%) | 32–<34 n/N (%) | 34–<35 n/N (%) | 35–<36 n/N (%) | 36–<37 n/N (%) | >37 n/N (%) |
|---|---|---|---|---|---|---|---|---|---|
| Gestational age<28 | 65 | 5/65 (7.7) | 20/65 (30.8) | 14/65 (21.5) | 16/65 (24.6) | 4/65 (6.2) | 4/65 (6.2) | 0 | 2/65 (3.1) |
| Gestational age 28–31 | 202 | – | 7/202 (3.4) | 87/202 (43.1) | 90/202 (44.5) | 8/202 (4.0) | 4/202 (2.0) | 4/202 (2.0) | 2/202 (1.0) |
| Gestational age 32–34 | 427 | – | – | – | 159/427 (37.2) | 204/427 (47.8) | 60/427 (14.1) | 1/427 (0.2) | 3/427 (0.7) |
| Gestational age 35–36 | 273 | – | – | – | – | – | 136/273 (49.8) | 124/273 (45.4) | 13/273 (4.8) |
| Total | 967 | 5/967 (0.5) | 27/967 (2.8) | 100/967 (10.3) | 263/967 (27.2) | 209/967 (21.6) | 205/967 (21.2) | 128/967 (13.2) | 30/967 (3.1) |
| **Accummulated (%)** | | | | | | | | | |
| Gestational age<28 | 65 | 7.7 | 38.5 | 60.0 | 84.6 | 90.8 | 97.0 | 97.0 | 100 |
| Gestational age 28–31 | 202 | | 3.4 | 46.5 | 91.0 | 95.0 | 97.0 | 99.0 | 100 |
| Gestational age 32–34 | 427 | | | | 37.2 | 85.0 | 99.1 | 99.3 | 100 |
| Gestational age 35–36 | 273 | | | | | | 49.8 | 95.2 | 100 |

PMA=postmenstrual age.

experience were not significantly associated with PMA at first breastfeeding. Adjusting for all the potentially delaying factors did not change the result (Table 4).

## Best performance at the first breastfeeding attempt

All infants had skin-to-skin contact before the first breastfeeding attempt. At the first breastfeeding attempt, 23.6% of the preterm infants would just "Smell the breast". This behaviour was almost equally distributed among the four GA groups except for the very preterm infants (Table 5).

Another 22.9% would "Lick and taste the milk" with a significant linear-by-linear association: decreasing frequency in increasing GA groups. The most frequent behaviour at the first breastfeeding attempt was "Seeks and finds breast, gets nipple in mouth" (31.6%) with no differences between GA groups. Significantly more of the very preterm infants did "Suckle and swallow briefly" compared to the other GA groups. The very few infants (2.6%) who displayed a breastfeeding behaviour beyond this at the first breastfeeding attempt were predominantly moderate and late preterm infants (GA 32–36). When infant performance was analysed in PMA groups, the step "Licks and tastes" was more frequent the lower

**Table 4. Factors associated with postmenstrual age at first breastfeeding attempt.**

| Explanatory variables | n/N (%) | Unadjusted analysis | | Adjusted for GA* | | Model adjusted for all | |
|---|---|---|---|---|---|---|---|
| | | Mean days (95% CI) | p-value | Mean days (95% CI) | p-value | Mean days (95% CI) | p-value |
| Gestational age, weeks | 838 | 3.86 (3.65 to 4.07) | <0.0001 | | | 3.97 (3.71 to 4.22) | <0.0001 |
| Respiratory support: | | | | | | | |
| Mechanical ventilation | 58/744 (7.8) | −11.56 (−15.29 to −7.84) | <0.0001 | 6.93 (4.20 to 9.65) | <0.0001 | 6.96 (4.20 to 9.71) | <0.0001 |
| Nasal-CPAP treatment or HFNC | 463/744 (62.2) | −10.63 (−12.69 to −8.57) | <0.0001 | −0.97 (−2.46 to 0.52) | 0.202 | −0.93 (−2.46 to 0.60) | 0.232 |
| Small for gestational age | 140/830 (16.9) | 1.96 (−0.49 to 4.41) | 0.117 | 0.83 (−0.71 to 2.37) | 0.288 | 0.50 (−1.18 to 2.18) | 0.561 |
| Not breastfed before | 582/827 (70.4) | 0.04 (−1.98 to 2.06) | 0.969 | 0.14 (−1.13 to 1.41) | 0.827 | 0.08 (−1.28 to 1.47) | 0.905 |

CPAP = Continuous positive airway pressure, HFNC = High-flow nasal cannula.

*N = 732 in final model.

**Table 5. Infants' best breastfeeding performance at first breastfeeding attempt across gestational age groups.**

| | Total population | GA<28 | GA 28–31 | GA 32–34 | GA 35–36 |
|---|---|---|---|---|---|
| | n/N (%) | n/N (%) | n/N (%) | n/N (%) | n/N (%) |
| Smells the breast | 234/992 (23.6) | 16/69 (23.2) | 32/210 (15.2)* | 115/440 (26.1) | 71/273 (26.0) |
| Licks and tastes the milk | 227/992 (22.9) | 25/69 (36.2)** | 62/210 (29.5)** | 95/440 (21.6)** | 45/273 (16.5)** |
| Seeks and finds breast, gets nipple in mouth | 313/992 (31.6) | 19/69 (27.5) | 60/210 (28.6) | 142/440 (32.3) | 92/273 (33.7) |
| Suckles and swallows briefly | 192/992 (19.3) | 9/69 (13.0) | 55/210 (26.2)* | 78/440 (17.7) | 50/273 (18.3) |
| Breastfeeds more effectively, reduced amount through feeding-tube/cup | 19/992 (1.9) | 0*** | 0*** | 5/440 (1.1)*** | 14/273 (5.1)*** |
| Breastfeeds majority of the feed | 2/992 (0.2) | 0 | 0 | 2/440 (0.5) | 0 |
| Breastfeeds a complete feed | 5/992 (0.5) | 0 | 1/210 (0.5) | 3/440 (0.7) | 1/273 (0.4) |

*Pearson Chi-Square <0.01, **Linear-by-Linear Association <0.0001, ***Pearson Chi-Square <0.01 GA<32 weeks vs. older.

GA= gestational age, CPAP = nasal Continuous Positive Airway Pressure.

the PMA, and the step "Breastfeeds more effectively" at the first attempt was more frequent with higher PMA corresponding to the findings in the GA groups (Supporting information, S3 Table).

## Nasal-CPAP

Of the 617 infants treated with nasal-CPAP, almost half of them (292) still had nasal-CPAP during the first breastfeeding attempt. The most frequent performance in infants maintained on and those weaned from nasal-CPAP was "seeks and finds the breast, gets nipple in mouth". There were no significant differences in breastfeeding performance among preterm infants with and without nasal-CPAP during first breastfeeding attempt, except for the step "Smells the breast", which was more frequent among infants with nasal-CPAP (25.7% vs. 17.5%, p = 0.014), (Table 6). Dichotomizing the Milky Way into behaviour with swallowing or not, did not reveal significant differences in percentage of infants swallowing between infants maintained on nasal-CPAP (63/292, 21.6%) and those weaned from nasal-CPAP (86/325, 26.5%), p = 0.157.

**Table 6. Infants' best breastfeeding performance at first breastfeeding attempt with or without nasal-CPAP.**

| Best performance of 617 preterm infants | Weaned from CPAP | At breast with CPAP | Pearson Chi-Square |
|---|---|---|---|
| | n/N (%) | n/N (%) | |
| *Behaviour without swallowing:* | | | |
| Smells the breast | 57/325 (17.5) | 75/292 (25.7) | 0.014 |
| Licks and tastes the milk | 82/325 (25.2) | 75/292 (25.7) | 0.897 |
| Seeks and finds breast, gets nipple in mouth | 100/325 (30.8) | 79/292 (27.1) | 0.310 |
| *Behaviour with swallowing:* | | | |
| Suckles and swallows briefly | 77/325 (23.7) | 57/292 (19.5) | 0.210 |
| Breastfeeds more effectively, reduced amount through feeding-tube/cup | 5/325 (1.5) | 5/292 (1.7) | 0.864 |
| Breastfeeds majority of the feed | 2/325 (0.6) | 0/292 (0) | 0.179 |
| Breastfeeds a complete feed | 2/325 (0.6) | 1/292 (0.3) | 0.627 |

CPAP = Continuous positive airway pressure.

## Establishment of exclusive breastfeeding

Of the 829 infants with data on both PMA at first breastfeeding attempt and use of bottle-feeding, a total of 95.7% (793/829) had the first oral experience at breast, 64.3% (533/829) established exclusively breastfeeding without introducing a bottle in the NICU and 31.4% (260/829) were introduced to the breast before the bottle.

Exclusive breastfeeding at the breast was established in 65.2% (570/874) of the infants (data missing for 118 infants). There was no correlation between PMA at first breastfeeding attempt and PMA at establishment of exclusive breastfeeding in all infants (Pearson's correlation 0.002 (p = 0,970), as well as no correlation among infants with GA < 32 weeks (Pearson's correlation −0.022 (p = 0.812)), see Supporting Information Scatterplot S2 Fig. Within each GA group there was no association between PMA at first breastfeeding attempt and successful establishment of exclusive breastfeeding (Supporting information, S5 Table).

## Discussion

In this large cohort of preterm infants, we found remarkably low PMA at first breastfeeding attempt in infants born before 28 gestational weeks. The only factor, except for higher GA, which delayed the timing was mechanical ventilation. In all GA groups infants predominantly performed on the first steps of the Milky Way without swallowing. Overall, first breastfeeding attempt while maintained on nasal-CPAP did not affect the breastfeeding performance.

### Timing of first breastfeeding attempt

Infants in our cohort had their first breastfeeding attempt at a lower PMA (from 27.57 weeks) compared with preterm infants in most studies from Sweden, the US and Canada, where the lowest PMA ranged between 27.9 and 31.0 weeks at first breastfeeding attempt [4,6,7,9]. The differences could be explained by local or national care protocols restricting early breastfeeding initiation in some of these countries [14,15].

Previously a systematic review established that exposure to the breast before 32 weeks' PMA is safe in stable preterm infants. Additionally, infants maintain oxygen saturation and increase body temperature during breastfeeding, in contrast to bottle-feeding [8]. Furthermore, breastfeeding does not seem to expend energy needed for growth in preterm infants [27]. In present study, no severe adverse events during breastfeeding of preterm infants were reported. This highlights the need to revise local and national guidelines to incorporate recommendations from the Neo-BFHI and WHO, suggesting that stable preterm infants should have unrestricted access to breastfeeding, regardless of GA, PMA, PNA or current weight [1–3]. Implementation should include training of NICU nurses in readiness for early breastfeeding.

### Factors associated with PMA at first breastfeeding attempt

We found that timing of first breastfeeding attempt was delayed by mechanical ventilation, which is expected as mechanical ventilation in Denmark is used for infants with severe conditions, who, in turn, are not stable and not ready to initiate early breastfeeding. In Denmark, nasal-CPAP is the first choice of respiratory support in preterm infants [21]. Treatment with nasal-CPAP was not associated with delay of first breastfeeding attempt in present study.

### Performance at first breastfeeding attempt

The first steps in the Milky Way were the most frequently reported, with an almost equal distribution across all GA groups. The infants showed interest in the breast by smelling, licking, tasting, seeking, and getting the nipple in the mouth. This performance could be interpreted as familiarization with the breast in preterm infants [28] and indicate that preterm infants, as term infants, need time to familiarize with the breast before starting sucking [16]. In the present study, approximately 60% of infants did not suck and 80% did not swallow at the first breastfeeding attempt. In contrast, in a Swedish population of healthy preterm infants, the infants had a higher PMA and more of them sucked and swallowed at first breastfeeding attempt [4]. It was, however, a selected population of healthy preterm infants. The differences may indicate that an earlier timing of the first breastfeeding attempt could lead to familiarization behaviours while at the breast. Breastfeeding behaviour at the first breastfeeding attempt beyond familiarization and briefly sucking was almost only seen in infants born from 32 gestational weeks, who are more mature, still this performance was rare. As we did not measure the single infant's development of breastfeeding competencies over time, we are not able to describe the pace in the development. Oral feeding is well-studied in bottle-fed preterm infants [13] but sucking skills might differ between breastfeeding and bottle-feeding, and with and without nasal-CPAP, and therefore research in bottle-feeding is not necessarily applicable to breastfeeding [29]. A US study showed in 1988 that preterm infants were more stable during breastfeeding than bottle-feeding [30].

### Nasal-CPAP

In present study, 29% of the infants had the first breastfeeding attempt while maintained on nasal-CPAP including 84% of the extremely preterm infants. A previous national Danish study from 2009–2010, found that 21% of all and 62% of the extremely preterm infants had the first breastfeeding attempt with nasal-CPAP, indicating an increase of initiating breastfeeding while maintained on nasal-CPAP [5]. Breastfeeding with nasal-CPAP has also been reported in Canada, the US, and Ireland [9,31–33], but has not yet been commonly referenced in Swedish and German studies [4,34].

To our knowledge, this is the first study to examine differences in performance at first breastfeeding attempt between infants weaned from and maintained on nasal-CPAP. Previously, PMA at first breastfeeding was assessed in a Canadian study including infants maintained on nasal-CPAP, but their performance was not described [9]. Remaining studies of breastfeeding and nasal-CPAP included infants after 32–34 weeks PMA, and PMA or performance at first breastfeeding were not reported [31,33].

For those infants only reaching the initial step "smell the breast", significantly more were maintained on nasal-CPAP (25%) than weaned from CPAP (17%) indicating slightly less energy when needing nasal-CPAP. In contrast, we found no differences in the other steps, and no differences in infants with swallowing behavior. So overall nasal-CPAP during first breastfeeding attempt was not associated with breastfeeding performance. Thus, we suggest with caution that breastfeeding performance at first breastfeeding attempt does not differ for stable preterm infants with and without nasal-CPAP.

### Establishment of exclusive breastfeeding

We found no correlation between PMA at first breastfeeding attempt and PMA at establishment of exclusive breastfeeding. Moreover, we found no association between PMA at first breastfeeding attempt and exclusive breastfeeding at discharge

in this cohort which could be explained by the high level of breastfeeding support to all dyads. The missing data for 12% of the infants could potentially contribute to the lack of significant differences but missing data seems randomly distributed (Supporting information, S4 Fig), and the observed differences very small (Supporting information, S5 Table). The implications of early breastfeeding attempts may result in more infants having their first oral experience at breast instead of bottle, more breastfeeds during NICU stay, higher maternal milk supply, higher breastfeeding exclusivity at discharge, and more content mothers [6,35]. A US study has shown that direct breastfeeding contributes to the mother's sustained milk supply in pumping mothers, and that the numbers of direct breastfeeds were associated with breastmilk feeding at discharge [6]. Also, mothers of preterm infants could benefit from early breastfeeding, including making a step towards normality and feeling important and connected to her baby [35]. Further, it is important for the mothers to experience mutual positive responses between her and her baby in order to have a positive breastfeeding experience [36], which will require a sensitive, respectful, and appropriate timing of support from the health care professionals. Early breastfeeding initiation would be a paradigm shift in many NICUs and require careful education and implementation.

About 96% of the infants in our cohort were introduced to the breast before (or completely without) being introduced to a bottle, which could also contribute to the high breastfeeding rate at discharge. Preterm infants in a US study who performed the first oral feed at breast were eight times more likely to be discharged to home receiving breast milk compared to those performing first oral feed with a bottle [7]. Our exclusive breastfeeding rate in preterm infants at discharge (65%) is high compared to other international studies (7% – 55%) [6,37–39], although most of these international studies do not report the timing of the first breastfeed.

Term infants need the first hour or more to familiarize with the breast during skin-to-skin contact, and so do preterm infants; in fact, due to immaturity, the familiarization period is expected to be much longer for preterm infants [28]. The preterm infants' need for familiarization should be respected without interfering, e.g., by offering a nipple shield or giving hands-on help. A Danish study found that 17% of preterm infants were offered a nipple shield at the day of first breastfeeding attempt, which does not support access needed for familiarization with the breast [40]. Furthermore, nipple shield use does not accelerate establishment of exclusive breastfeeding [5]. Instead, it has been found to double the risk of not breastfeeding exclusively at discharge [25], and this risk is also significantly increased if the motive is "Infant falling asleep at the breast" [40]. This suggests that nipple shield use is not the right solution for preterm infants showing familiarization cues at the breast. Instead, nurses should be patient and support the mother in observing the infants' familiarization cues and accept this takes time.

## Strengths and limitations

The strengths of the present study are the multicentre design, the inclusion of preterm infants from 23 to 36 weeks, and the large numbers of participants with data from 79.8% of those who consented to participate. In survey studies, a response rate above 60% is deemed acceptable [41]. Further, we also have unique results.

Data on infant performance at first breastfeeding attempt was solely assessed by the mothers. However, Nyqvist et al. found a high interrater reliability between nurses and mothers' observations of preterm infants breastfeeding performance [4], which justifies why mothers' observations are considered valid. Information of the first breastfeeding attempt (PMA, performance, and nasal-CPAP) had a maternal recall period of median nine days for 88% and 52 days for 12% of the mothers, respectively. As the first breastfeeding attempt is an important milestone for preterm infants, and since maternal recall of breastfeeding duration has been found valid for a six-year period [42], recall bias should not influence any differences in the analyses of delayed initiation, performance, or nasal-CPAP.

It was not possible to answer "no reaction" as a response to performance at the first breastfeeding attempt, but the step "Smells the breast" could also be interpreted as "mouth/nose against nipple" which might not include reactions from the infant. Another limitation is that details on infant treatment that could delay breastfeeding initiation were limited to respiratory support. However, as mechanical ventilation is only used for very challenged preterm infants in

Denmark, we do believe that the influence of sicker infants was captured in that variable. We had no information of the infants' medical conditions at first breastfeeding attempt or the physical reactions during the first breastfeeding attempt, e.g., desaturations, bradycardia. In Denmark, severe adverse events are systematically reported to learn and prevent similar situations. Our study was led by the Knowledge Centre for Breastfeeding Infants with Special Needs. No severe adverse events during breastfeeding of preterm infants have been reported to the Knowledge Centre during the study period.

The definition of infant stability related to breastfeeding was "Infants who respond to routine care and handling without experiencing severe apnoea, desaturation and bradycardia" [1–3]. This definition could, in practice, vary widely. As stability was not clearly defined in the NICUs' protocols, it could be interpreted differently between and within participating NICUs. Further, the interpretation of stability could vary depending on the health professionals experience with early breastfeeding and could limit implementation of new practice based on the present results. WHO suggests NICUs not to be unreasonably restrictive and to "set both realistic and ambitious targets" for early breastfeeding in preterm infants [3]. The criterium for breastfeeding initiation in all Danish NICUs was "stable infant" and not a certain PMA or weight. Thus, the present study was not designed to determine whether stable infants restricted from early breastfeeding initiation are delayed in timing of the establishment of exclusive breastfeeding or are less breastfed. The study was conducted in 2016–2019, and the national recommendation to initiate breastfeeding when the preterm infant is stable has not changed. Some mother-infant pairs were for different reasons not included in the study for whom we had no data. Thus, the results of our study could only be generalized to similar populations.

## Conclusion

The majority of the extremely and very preterm infants in this cohort had their first breastfeeding attempt before 32 weeks PMA. In all GA groups infants predominantly performed on the first steps of the Milky Way without swallowing. When breastfeeding was initiated as soon as the infants were cardiorespiratory stable as recommended in Neo-BFHI and by WHO, they familiarized with the breast, and in general did not suck the first time they were presented to the breast. And this pattern seemed not affected by GA, PMA or nasal-CPAP treatment during first breastfeeding attempt. Mechanical ventilation was found to delay first breastfeeding attempt but having nasal-CPAP was not. Hence, nasal-CPAP should not be a barrier to breastfeeding initiation in stable infants based on WHO's definition. We found no correlation between PMA at first breastfeeding attempt and PMA at establishment of exclusive breastfeeding, nor exclusive breastfeeding at discharge. Early initiation of breastfeeding seems possible in stable preterm infants. Guidelines and policies in NICUs should comply with the recommendations by Neo-BFHI and WHO. Based on our findings, NICUs where breastfeeding initiation is restricted by a fixed PMA may be inspired to follow WHO's recommendations and set realistic and ambitious targets for early breastfeeding initiation in preterm infants and thus benefit preterm infants and their mothers by allowing more time for breastfeeding opportunities. Future research should include preterm infants' physical reactions during first breastfeed attempt with and without nasal-CPAP and their breastfeeding development through the Nine Stages skin-to-skin.

## Supporting information

**S1 Table. First breastfeeding attempt with or without nasal-CPAP across gestational age groups.**
(DOCX)

**S2 Fig. Scatter Plot of postmenstrual age at first breastfeeding attempt by gestational age.**
(DOCX)

**S3 Table. Infants' best breastfeeding performance at first attempt across postmenstrual age groups.**
(DOCX)

**S4 Fig. Scatter Plot of postmenstrual age at establishment of exclusive breastfeeding by postmenstrual age at first breastfeeding attempt.**
(DOCX)

**S5 Table. Mean differences in days of postmenstrual age at first breastfeeding attempt between infants who did and did not establish exclusive breastfeeding.**
(DOCX)

## Acknowledgments

We want to thank the nurse managers and contact persons in the participating Danish NICUs for supporting the study and enrolling participants, the participating mothers for contributing data to the study, and the National Expert Panel in Breast-feeding Infants with Special Needs for contributing to the development of the study.

## Author contributions

**Conceptualization:** Ragnhild Maastrup, Hanne Kronborg, Helle B. Sandfeld.

**Data curation:** Ragnhild Maastrup, Sisse Walloee.

**Formal analysis:** Ragnhild Maastrup, Ane L. Rom.

**Funding acquisition:** Ragnhild Maastrup.

**Project administration:** Ragnhild Maastrup.

**Resources:** Ragnhild Maastrup, Helle B. Sandfeld.

**Supervision:** Hanne Kronborg.

**Validation:** Sisse Walloee.

**Writing – original draft:** Ragnhild Maastrup, Ane L. Rom.

**Writing – review & editing:** Ragnhild Maastrup, Sisse Walloee, Hanne Kronborg, Helle B. Sandfeld, Ane L. Rom.

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
