## [Decision Letter · Decision Letter 0]

Dear Dr. Maastrup,

Thank you for submitting your manuscript to PLOS ONE. After careful consideration, we feel that it has merit but does not fully meet PLOS ONE’s publication criteria as it currently stands. Therefore, we invite you to submit a revised version of the manuscript that addresses the points raised during the review process.

We look forward to receiving your revised manuscript.

Kind regards,

Abdelaziz Hendy, PHD

Academic Editor

PLOS ONE

Journal Requirements: When submitting your revision, we need you to address these additional requirements. 1. Please ensure that your manuscript meets PLOS ONE's style requirements, including those for file naming. The PLOS ONE style templates can be found at https://journals.plos.org/plosone/s/file?id=wjVg/PLOSOne_formatting_sample_main_body.pdf and https://journals.plos.org/plosone/s/file?id=ba62/PLOSOne_formatting_sample_title_authors_affiliations.pdf

Reviewers' comments:

Reviewer's Responses to Questions

**Comments to the Author**

1. Is the manuscript technically sound, and do the data support the conclusions?

Reviewer #1: Yes

2. Has the statistical analysis been performed appropriately and rigorously?

Reviewer #1: No

3. Have the authors made all data underlying the findings in their manuscript fully available?

Reviewer #1: Yes

4. Is the manuscript presented in an intelligible fashion and written in standard English?

Reviewer #1: Yes

Reviewer #1: Reviewer Report Preterm infants’ first breastfeeding attempt: Early initiation and performance with or without nasal-CPAP. A large cohort study

This study is a well-written. It addresses an important issue concerning Early initiation and performance of breastfeeding with or without nasal CPAP

The title is adequately addressed.

There are some comments to help improve the manuscript.

Abstract:

- The abstract provides a brief overview of the study objectives, methods, and key findings. However, it would benefit from including key results with a p-value, which would help the reader understand and make a clearer statement of the study's significance. Additionally, the abstract could mention the potential implications of the findings for clinical practice or patient outcomes.

- Please check the manuscript for any grammar mistakes. Some grammar errors require correction, such as

In line 19 (it should be Baby Friendly instead of Baby- friendly)

In line 40 (put coma, after (in 2020)

Main manuscript:

- The introduction provides a comprehensive background on Early initiation and the importance of breastfeeding in preterm infants with or without nasal-CPAP. The gap is clearly addressed.

- However, it would be helpful to organize the introduction's hierarchical structure and flow and provide contrasting evidence or practices that highlight the research gap or problem your study aims to address.

________

Methods: Study design, setting, and participants

- Design should be clearly identified as a “Retrospective multisite observational cohort study”.

- The study mentioned 13 Danish Neonatal Intensive Care Units, and the second line mentioned all 17 Danish Neonatal Intensive Care Units. It would be helpful to clarify whether those numbers are the same.

- Assuming that you selected 13 out of 17 NICUs, but the methodology does not clarify why only 13 of 17 Danish NICUs participated, it would be better to clarify the reasons for non-participation in order to clarify potential biases or limitations in the sample.

- in the design and setting, lines 95- 107 contain a lot of information about the Danish health care system. It would be better to put this information in the introduction.

- In the data collection, it was mentioned that the questionnaires were adapted from a previous study but did not provide specific information regarding the aspects that were modified or revised. This is essential for evaluating the accuracy and dependability of the instruments employed.

- The data were collected from the mothers of preterm infants regarding breastfeeding attempts… If breastfeeding attempts are solely based on maternal reports, recall bias may exist, especially since some data was collected well after events occurred.

_____

Results:

- In the table 1 …..You’ve reported that the preterm(small for gestational age ) was 182/982, and the total number was 992 .. Clarify whether this reflects a subgroup analysis or if there are missing data points for some infants in these categories.

- It was reported that Nasal-CPAP or high-flow nasal cannula (not MV) and Mechanical Ventilation (MV) with different denominators (876 for both), suggesting that not all of the 992 infants are considered in these categories. Clarify whether this reflects a subgroup analysis or if there are missing data points for some infants in these categories.

- It is mentioned in the table comments that 7.9% were mechanically ventilated, followed by nasal-CPAP in all but three infants. Is that reported in the table ?

- It is reported that the mother's characteristics (e.g., Breastfed before 846, Mode of delivery 848) remain inconsistent. Clarify If these figures are supposed to represent subsets due to missing data, it should be explicitly stated.

- It was mentioned that 43/838 for smoking and educational level 838 ……which is inconsistent between smoking and educational level… clarify if this implies missing data if the total is supposed to be 858.

- In Table 2 ……..It was reported that the total number across all Gestational age groups included for analysis was (968), but it sounds inconsistent because the sum of infants in all GA groups (65 + 202 + 427 + 273 = 967) totals 967, not 968 as mentioned

- In Table 3…..can you check the accumulated percentage of the number of infants At <28 weeks was 66 or 65 ?

________

Discussion: provides a comprehensive interpretation of the study findings and relates them to the existing literature. When comparing with other studies, we can focus on the differences in care protocols or the sociocultural context that could potentially impact breastfeeding practices. This would bolster the argument regarding the uniqueness or applicability of your findings.

The conclusion provides a concise summary of the study's main findings. However, it could be strengthened by emphasizing the practical implications of the results( how change practice in NICUs where breastfeeding initiation is restricted by a fixed PMA and thus benefit preterm infants and their mothers) and suggesting avenues for future research

Reference:

- When possible, include more recent studies; older studies, unless they are justified historically, may not reflect the most recent research or guidelines.

**Do you want your identity to be public for this peer review?** For information about this choice, including consent withdrawal, please see our Privacy Policy

Reviewer #1: No

---

## [Author Response · Author response to Decision Letter 1]

11 Oct 2024

Dear reviwer.

We thank you for the opportunity to respond to the valuable comments and suggestions for our manuscript.

We appreciate the constructive feedback provided, which allow us to clarify important aspects of our study.

Hopefully the manuscript is now acceptable for publication.

In the uploaded file, we have responded to every comment.

All line numbers refer to the version with tracked changes.

---

## [Decision Letter · Decision Letter 1]

Dear Sir Maastrup,

Thank you for submitting your manuscript to PLOS ONE. After careful consideration, we feel that it has merit but does not fully meet PLOS ONE’s publication criteria as it currently stands. Therefore, we invite you to submit a revised version of the manuscript that addresses the points raised during the review process.

Kind regards,

Edison Arwanire Mworozi, M.D

Academic Editor

PLOS ONE

Journal Requirements:

Additional Editor Comments:

tHE COMMENTS MADE BY REVIEWERS ARE CRITICAL AND NEED TO BE ADDRESSED.

Reviewers' comments:

Reviewer's Responses to Questions

**Comments to the Author**

Reviewer #2: All comments have been addressed

Reviewer #3: All comments have been addressed

2. Is the manuscript technically sound, and do the data support the conclusions?

Reviewer #2: No

Reviewer #3: Yes

3. Has the statistical analysis been performed appropriately and rigorously?

Reviewer #2: No

Reviewer #3: Yes

4. Have the authors made all data underlying the findings in their manuscript fully available?

Reviewer #2: No

Reviewer #3: Yes

5. Is the manuscript presented in an intelligible fashion and written in standard English?

Reviewer #2: Yes

Reviewer #3: Yes

Reviewer #2: Thank you very much for inviting me to review this article. I read with interest the article by Ragnhild Maastrup and colleagues which analyzed a questionnaire, adapted from their previous study, regarding the performance of preterm infants during breast feeding.

The authors aimed to analyze their previous questionnaire from a different perspective focusing on PMA at first attempt of breast-feeding, factors delaying the first attempt, performance at first attempt with and without nasal-CPAP, and the correlation between PMA at first attempt and PMA at exclusive breastfeeding establishment.

However, the study did not answer the raised questions by the authors as it mainly depended on maternal responses on the performance of their infants during feeding (which is very subjective) rather than a truly structured objective assessment. The data presented are mainly descriptive and lacked a comparative analysis between infants who had early initiation versus those who had late initiation to better understand the obstacles and appreciate the outcomes.

The following points need to be addressed by the authors.

1. The title (Preterm infants’ first breastfeeding attempt: Early initiation and performance with or without nasal-CPAP. A large cohort study) doesn't reflect the design or the methodology of the study. It gives the impression that the design is a comparative analysis between infants initiating breast feeding while on n-CPAP versus those without n-CPAP. It doesn't mention that this is a questionnaire study or a retrospective design. Please follow STROBE stands for reporting observational trials.

2. What was the criteria for initiation of breast-feeding attempt? Is there a policy for that? Did all the units participated in the study follow the same policy for initiation of attempts?

3. If 551 preterm infants were on n-CPAP or HFNC, and 292 (53%) of them received first breast feeding attempt while on CPAP. What controlled infant selection and feeding practice in the participating units?

4. The question of breast-feeding performance was mentioned twice in questionnaire 1 (question number 20) and in questionnaire 2 (question number 4) with the exact same items in the original study. As stated by the authors, the mother received a link to the first questionnaire approximately one week after delivery, and a link to the second at the infant’s final discharge to home. Which response was counted in these results considering that mothers in the second questionnaire has more experience and are more able to retrospectively interpret her infant performance during first attempt in different way? Was there any difference between mother responses in the first and second questionnaires?

5. In the original study the questionnaires were sent to two groups of mothers, the control group (before training of the nurses) and the intervention group (after training of the nurses). Was there any difference between the two groups?

6. The first item in the performance at first feeding attempt was "Smells the breast", What was the infant reaction which gave the mother this impression? How was the mother able to conclude that her infant smelled the breast particularly in the group of infants on n-CPAP with the interface on?

7. Is there any chance that the infant did not do any reaction or response while on the breast? This option was not given to the mother in the original questionnaire. What are the chances that mothers have chosen the option of "Smell the breast" when their infants had no response as an alternative response?

8. Although the authors, were interested in assessing the value of early initiation of breast feeding (PMA at first attempt) on the later ability of full breast feeding (PMA at exclusive breastfeeding establishment), they included 273 (27.5%) infants of 35 to 36 weeks' gestation who have inherited ability to establish breast feeding and will, probably, not benefit from early intervention. I believe this is the reason that the authors did not find a significant correlation between PMA at first breastfeeding attempt and PMA at establishment of exclusive breastfeeding (Pearson's correlation 0.002 (p=0,970). A subgroup correlation analysis for infants less than 32 weeks' gestation in whom early intervention is expected to make a difference may have a different significance.

9. The results of Table 2 are very expected and too obvious to be compared statistically. 35- and 36-weeks infants are usually born to the breast directly, what is the value of comparing their PNA and weight at first feeding attempt to a preterm infant less than 28 weeks' gestation?

10. Table 4 looked only on gestational age, respiratory support, SGA, and previous maternal experience with breast feeding as factors that may delay initiation of breast feeding. However, I believe there are other many factors which are of importance including (and not limited to) neonatal sepsis, inotropes, PDA and PDA treatment, etc.

11. Table 5 the number of infants with a response of "Smell the breast" to the total population should be 234/992 and not 243/992.

Reviewer #3: Thank you for asking me to review this revised manuscript. \i think the manuscript is nicely presented. I this there is a few points which needs some clarity.

The first point is related to mechanical ventilation, both direct and indirect ventilation. It is a bit confusing how this data is being presented. in the smaller infants, <28 weeks and 28-31 weeks, it is not clear in those groups who were still on NCAP or HF at the time of the first BF attempt. Also what steps were taken in the NICUs for babies to be given the go ahead with respect to first attempts. Were volumes, as determined by test weighing, logged. This may be difficult to answer as the data is based on maternal questionnaires.

The second point is related to the questionnaires completed by the mothers. There seems to be very large window for completion. Is there an explanation and what steps were in place regarding getting the questionnaires?

**Do you want your identity to be public for this peer review?** For information about this choice, including consent withdrawal, please see our Privacy Policy

Reviewer #2: **Yes: ** Nehad Nasef

Reviewer #3: No

---

## [Author Response · Author response to Decision Letter 2]

7 Jan 2025

Please see attached file in which we have responded point to point to the reviewers’ comments.

---

## [Decision Letter · Decision Letter 2]

Dear Dr. Maastrup,

 If applicable, we recommend that you deposit your laboratory protocols in protocols.io to enhance the reproducibility of your results. Protocols.io assigns your protocol its own identifier (DOI) so that it can be cited independently in the future. For instructions see: https://journals.plos.org/plosone/s/submission-guidelines#loc-laboratory-protocols. Additionally, PLOS ONE offers an option for publishing peer-reviewed Lab Protocol articles, which describe protocols hosted on protocols.io. Read more information on sharing protocols at https://plos.org/protocols?utm_medium=editorial-email&utm_source=authorletters&utm_campaign=protocols.

Kind regards,

Edison Arwanire Mworozi, M.D

Academic Editor

PLOS ONE

Additional Editor Comments:

Please address the second reviewers comments.

Reviewers' comments:

Reviewer's Responses to Questions

**Comments to the Author**

Reviewer #2: All comments have been addressed

Reviewer #3: (No Response)

2. Is the manuscript technically sound, and do the data support the conclusions?

Reviewer #2: Yes

Reviewer #3: Partly

3. Has the statistical analysis been performed appropriately and rigorously?

Reviewer #2: Yes

Reviewer #3: Yes

4. Have the authors made all data underlying the findings in their manuscript fully available?

Reviewer #2: Yes

Reviewer #3: Yes

5. Is the manuscript presented in an intelligible fashion and written in standard English?

Reviewer #2: Yes

Reviewer #3: Yes

Reviewer #2: I would like to thank the authors for their detailed responses and adequately addressing all comments

Reviewer #3: Thank you for asking me to review the revised manuscript. I think that describing the association between first exposure and feeding at final discharge is extremely important.

I am not sure that you addressed the concerns expressed by the reviewers related to those infants who had their first exposure to the breast while on CPAP. The standard of care for the majority of preterm infants, especially now is to use less invasive ventilation to reduce barotrauma. So it is important to recognize this approach in care and not to minimize lung disease and immaturity. The findings as you describe in Table 1 shows that when babies are born after 30 weeks gestation, they do have the capacity to develop feeding skills but these babies are more mature. What is still not clear is how are babies born under 28 weeks determined to be stable enough to experience exposure. What medical criteria was in place to consider this. This was not addressed and I think that this is a major limitation and should be identified as such. Maternal recall may not be able to address this.

It is also important to note that you have a number of mothers who chose not to complete the questionnaire. This needs to be addressed in the paper. You also do not give any idea as to how babies were determined to be stable. That is a limitation because this paper is based strictly on maternal recall. You did not get any data about the babies directly from a medical perspective. From a neonatologist perspective, this is critical especially for the extremely immature preterm infant.

In the paper, you state that 1819 infants were eligible but only 1243 were included; why were 576 not included and what were the reasons for not including them. we should also know what was the distribution of the entire eligible infants and why they ended up being excluded. In total 992, which is just over 50%, were included. From a research perspective, this does not allow adequate information to determine generalizability.

Although I recognize the value of maternal input, very little is provided regarding medical stability. This was not addressed in the paper. Consequently little guidance is provided to the reader on how to implement steps for the very preterm infant. This again is a limitation. As authors, you were able to show it is possible but provided little more than that.

The title is misleading in that it suggests that you evaluated the role of CPAP more in depth which in fact you did not. Consequently, this needs to be addressed and is a limitation and the title needs to change.

**Do you want your identity to be public for this peer review?** For information about this choice, including consent withdrawal, please see our Privacy Policy

Reviewer #2: **Yes: ** Nehad Nasef

Reviewer #3: No

---

## [Author Response · Author response to Decision Letter 3]

25 Feb 2025

We have answered point to point to the reviewers’ comments in the attached file.

---

## [Decision Letter · Decision Letter 3]

Dear Dr. Maastrup,

Thank you for submitting your manuscript to PLOS ONE. After careful consideration, we feel that it has merit but does not fully meet PLOS ONE’s publication criteria as it currently stands. Therefore, we invite you to submit a revised version of the manuscript that addresses the points raised during the review process.

We look forward to receiving your revised manuscript.

Kind regards,

Mona Nabulsi, MD, MS

Academic Editor

PLOS ONE

Journal Requirements:

Reviewers' comments:

Reviewer's Responses to Questions

**Comments to the Author**

Reviewer #2: All comments have been addressed

Reviewer #4: (No Response)

2. Is the manuscript technically sound, and do the data support the conclusions?

Reviewer #2: Yes

Reviewer #4: Yes

3. Has the statistical analysis been performed appropriately and rigorously?

Reviewer #2: Yes

Reviewer #4: Yes

4. Have the authors made all data underlying the findings in their manuscript fully available?

Reviewer #2: Yes

Reviewer #4: Yes

5. Is the manuscript presented in an intelligible fashion and written in standard English?

Reviewer #2: Yes

Reviewer #4: Yes

Reviewer #2: I would like to thank the authors for the effort they made in answering reviewer's comments. Authors clearly addressed all reviewers' comments. I have no further comments to add.

Reviewer #4: The authors address an important topic and describe a practice that is not common which is allowing premature babies to attempt breastfeeding at a lower age than developmentally capable and while on respiratory support. In this observational muliticenter cohort study, it was noted that preterm newborns as early as 27 weeks attempted breastfeeding, the main reason to prevent such attempts per this study were patients being on mechanical ventilation. However this practice did not affect the rate of breastfeeding later on.

The authors have addressed the concerns previously raised by reviewers but I would still ask for some clarifications on the following matters:

- The study is from the period 2016-2019, which is now 6-9 years old, have their been any changes in practice or recommendations on that matter? Can the authors clarfiy that in the discussion section?

- For the second questionnaire, the recall time was 52 days which increases recall bias especially for mothers describing their first attempt and in order to plot it on the "milky way".

- The authors mention that babies had to be clnically stable in order to attempt breastfeeding, but there were no adverse events reported during these attempts. Was this monitored or part of the questionnaire? The authors should explicitly mention that, especially that this practice was not associated with successful establishment of exclusive breastfeeding. This would help the reader balance the benefit vs risk of incorporating this practice.

**Do you want your identity to be public for this peer review?** For information about this choice, including consent withdrawal, please see our Privacy Policy

Reviewer #2: **Yes: ** Nehad Nasef

Reviewer #4: No

---

## [Author Response · Author response to Decision Letter 4]

4 Jun 2025

Thank you for relevant comments and suggestions. We have answered point by point in the attached file.

---

## [Editor Report · Decision Letter 4]

Dear Dr. Maastrup,

Please submit your revised manuscript by Jul 28 2025 11:59PM, with a detailed letter explaining your response to the reviewer. If you will need more time than this to complete your revisions, please reply to this message or contact the journal office at plosone@plos.org . **A rebuttal letter that responds to each point raised by the academic editor and reviewer(s). You should upload this letter as a separate file labeled 'Response to Reviewers'.**A marked-up copy of your manuscript that highlights changes made to the original version. You should upload this as a separate file labeled 'Revised Manuscript with Track Changes'.An unmarked version of your revised paper without tracked changes. You should upload this as a separate file labeled 'Manuscript'.

We look forward to receiving your revised manuscript.

Kind regards,

Mona Nabulsi, MD, MS

Academic Editor

PLOS ONE
---

## [Author Response · Author response to Decision Letter 5]

13 Jun 2025

The point-to-point response to reviewers is attached in the end of the pdf in a table with the manuscript text in italic. We do also copy the comments and answers below in a simple text for-mat.

Answer to reviewer #4

Comment 1 from reviewer

The authors have addressed the concerns previously raised by reviewers but I would still ask for some clarifications on the following matters:

- The study is from the period 2016-2019, which is now 6-9 years old, have their been any changes in practice or recommendations on that matter? Can the authors clarfiy that in the discussion section?

Authors’ response to comment 1:

Thank you for the relevant suggestion. We have added line 439-441:

The study was conducted in 2016 – 2019, and the national recommendation on breastfeeding initia-tion among stable preterm infant has not changed.

Comment 2 from reviewer

- For the second questionnaire, the recall time was 52 days which increases recall bias especially for mothers describing their first attempt and in order to plot it on the "milky way".

Authors’ response to comment 2:

We acknowledge the comment from the reviewer and have already tried to address this in the limitation section line 414-418, where we describe that 12% had a recall period on 52 days (median) and 88% had a period of only nine days (median) of recall:

Information of the first breastfeeding attempt (PMA, performance, and nasal-CPAP) had a mater-nal recall period of median nine days for 88% and 52 days for 12% of the mothers, respectively. As the first breastfeeding attempt is an important milestone for preterm infants, and since maternal recall of breastfeeding duration has been found valid for a six-year period [42], recall bias should not influence any differences in the analyses of delayed initiation, performance, or nasal-CPAP.

To further clarify, that in Denmark ‘The Milky Way’ has been used since 2008 to inform parents and health care professionals about breastfeeding among preterm infants, suggesting that the mothers were likely to be familiar with the steps, we have now added the following to the methods section line 168-170:

The Milky Way has been used nationwide in Denmark since 2008 to inform parents and health care professionals about preterm breastfeeding [26]. Mothers of preterm infants were therefore likely to be familiar with the steps in the Milky Way.

New reference added: Danish Health Authority 2008. Breastfeeding – a handbook for health care professionals.

Comment 3 from reviewer

- The authors mention that babies had to be clnically stable in order to attempt breastfeeding, but there were no adverse events reported during these attempts. Was this monitored or part of the questionnaire? The authors should explicitly mention that, especially that this practice was not as-sociated with successful establishment of exclusive breastfeeding. This would help the reader bal-ance the benefit vs risk of incorporating this practice.

Authors’ response to comment 3:

Thank you for the possibility to further clarify.

Adverse events were not monitored or part of the questionnaire, which has been described as a limitation line 424-429:

We had no information of the infants’ medical conditions at first breastfeeding attempt or the physical reactions during the first breastfeeding attempt e.g., desaturations, bradycardia. In Den-mark, severe adverse events are systematically reported to learn and prevent similar situations. Our study was led by the Knowledge Centre for Breastfeeding Infants with Special Needs. No severe adverse events during breastfeeding of preterm infants have been reported to the Knowledge Cen-tre during the study period.

Further, we have already tried to explicitly mention that we did not find statistically significant asso-ciations between PMA at first breastfeeding attempt and exclusive breastfeeding at discharge, in the results section line 302-307:

There was no correlation between PMA at first breastfeeding attempt and PMA at establishment of exclusive breastfeeding in all infants (Pearson's correlation 0.002 (p=0,970), as well as no correlation among infants with GA<32 weeks (Pearson’s correlation -0.022 (p=0.812)), see Supporting Infor-mation Scatterplot Figure S4. Within each GA group there was no association between PMA at first breastfeeding attempt and successful establishment of exclusive breastfeeding (Supporting infor-mation, Table S5).

and in the discussion section line 372-374:

Moreover, we found no association between PMA at first breastfeeding attempt and exclusive breastfeeding at discharge in this cohort which could be explained by the high level of breastfeeding support to all dyads.

This is repeated in the conclusion line 452-454: We found no correlation between PMA at first breastfeeding attempt and PMA at establishment of exclusive breastfeeding, nor exclusive breast-feeding at discharge.

Infants in this study were not restricted from early breastfeeding initiation. To clarify the limitations of the design we added “or are less breastfed.“ in line 439.

Line 436-439: The criterium for breastfeeding initiation in all Danish NICUs was “stable infant” and not a certain PMA or weight. Thus, the present study was not designed to determine whether stable infants restricted from early breastfeeding initiation are delayed in timing of the establishment of exclusive breastfeeding or are less breastfed.

We hope the reviewer will find this appropriate. Otherwise please let us know.

---

## [Decision Letter · Decision Letter 5]

Preterm infants’ first breastfeeding attempt: Early initiation and performance. A large multicentre questionnaire study based on maternal observations.

PONE-D-24-15661R5

Dear Dr. Maastrup,

We’re pleased to inform you that your manuscript has been judged scientifically suitable for publication and will be formally accepted for publication once it meets all outstanding technical requirements.

Kind regards,

Mona Nabulsi, MD, MS

Academic Editor

PLOS ONE

Additional Editor Comments (optional):

Reviewers' comments:

Reviewer's Responses to Questions

**Comments to the Author**

Reviewer #4: (No Response)

2. Is the manuscript technically sound, and do the data support the conclusions?

Reviewer #4: (No Response)

3. Has the statistical analysis been performed appropriately and rigorously?

Reviewer #4: (No Response)

4. Have the authors made all data underlying the findings in their manuscript fully available?

Reviewer #4: (No Response)

5. Is the manuscript presented in an intelligible fashion and written in standard English?

Reviewer #4: (No Response)

Reviewer #4: (No Response)

**Do you want your identity to be public for this peer review?** For information about this choice, including consent withdrawal, please see our Privacy Policy

Reviewer #4: No

---

## [Editor Report · Acceptance letter]

PONE-D-24-15661R5

PLOS ONE

Dear Dr. Maastrup,

I'm pleased to inform you that your manuscript has been deemed suitable for publication in PLOS ONE. Congratulations! Your manuscript is now being handed over to our production team.

Kind regards,

on behalf of

Dr. Mona Nabulsi

Academic Editor

PLOS ONE